# Predictive Analysis of the Pro-Environmental Behaviour of College Students Using a Decision-Tree Model

**DOI:** 10.3390/ijerph19159407

**Published:** 2022-07-31

**Authors:** Qiaoling Wang, Ziyu Kou, Xiaodan Sun, Shanshan Wang, Xianjuan Wang, Hui Jing, Peiying Lin

**Affiliations:** 1Beijing Academy of Educational Sciences, Beijing 100036, China; wangqiaoling2021@foxmail.com (Q.W.); wxj6598@163.com (X.W.); 2College of Teacher Education, Capital Normal University, Beijing 100048, China; kouziyu@foxmail.com; 3Institute of Education, University College London, London WC1E 6BT, UK; xiaodan.sun.16@ucl.ac.uk; 4Department of Foreign Language, Guangdong University of Science & Technology, Dongguan 523070, China; wangshanshangdust@foxmail.com; 5National Center for Schooling Development Programme, Ministry of Education, Beijing 100032, China; jinghui@csdp.edu.cn

**Keywords:** decision-tree model, college student pro-environmental behaviour, predictive analysis

## Abstract

The emergence of the COVID-19 pandemic has hindered the achievement of the global Sustainable Development Goals (SDGs). Pro-environmental behaviour contributes to the achievement of the SDGs, and UNESCO considers college students as major contributors. There is a scarcity of research on college student pro-environmental behaviour and even less on the use of decision trees to predict pro-environmental behaviour. Therefore, this study aims to investigate the validity of applying a modified C5.0 decision-tree model to predict college student pro-environmental behaviour and to determine which variables can be used as predictors of such behaviour. To address these questions, 334 university students in Guangdong Province, China, completed a questionnaire that consisted of seven parts: the Perceived Behavioural Control Scale, the Social Identity Scale, the Innovative Behaviour Scale, the Sense of Place Scale, the Subjective Norms Scale, the Environmental Activism Scale, and the willingness to behave in an environmentally responsible manner scale. A modified C5.0 decision-tree model was also used to make predictions. The results showed that the main predictor variables for pro-environmental behaviour were willingness to behave in an environmentally responsible manner, innovative behaviour, and perceived behavioural control. The importance of willingness to behave in an environmentally responsible manner was 0.1562, the importance of innovative behaviour was 0.1404, and the perceived behavioural control was 0.1322. Secondly, there are 63.88% of those with high pro-environmental behaviour. Therefore, we conclude that the decision tree model is valid in predicting the pro-environmental behaviour of college student. The predictor variables for pro-environmental behaviour were, in order of importance: Willingness to behave in an environmentally responsible manner, Environmental Activism, Subjective Norms, Sense of Place, Innovative Behaviour, Social Identity, and Perceived Behavioural Control. This study establishes a link between machine learning and pro-environmental behaviour and broadens understanding of pro-environmental behaviour. It provides a research support with improving people’s sustainable development philosophy and behaviour.

## 1. Introduction

The COVID-19 pandemic has a catastrophic impact on the achievement of the sustainable development goals (SDGs) for 2030, and human development is at a critical moment in history [1]. To implement the 2030 Agenda for Sustainable Development, UNESCO is reorienting education towards sustainable development and recognize college students as major contributors to the achievement of the SDGs. College students face the negative impact of COVID-19 pandemic and climate changes, while their pro-environmental behaviour is crucial for mitigating these negative impacts and driving social transformation, economic growth, and environmental sustainability [2].

Existing meta-analytics related to pro-environmental behaviour suggest that environmental awareness, environmental education and social norms might predict pro-environmental behaviour [3]. Social norms have a positive impact on pro-environmental behaviour. A study on the relationship between environmental activism, environmental behaviour, and social identity among 131 university students in Australia revealed that the relationship between social identity and environmental activism is indirect [4]. Further, human self-affirmation can reduce the problem of household food waste and may also contribute to pro-environmental behaviour [5].

In addition to this, corporate environmental responsibility can influence employees to adopt pro-environmental behaviours at work and increase the likelihood that they will consider the environmental consequences of their personal actions [6]. In terms of sense of place, the psychosocial drivers of behaviour can lead to changes in pro-environmental behaviour on a regional scale [7]. The study applies boosting theory and normative theory of value beliefs to the sustainable management of play destinations. The greater the visitor’s sense of responsibility for the play destination, the greater the impact on pro-environmental behaviour [8]. At the scale of the private sphere of residents, guiding residents of a neighbourhood toward environmentally friendly living and improving the area’s private sewage system can promote pro-environmental behaviour [9]. However, few studies have focused on the predictive role of all these factors (attitude, lifestyle and sense of place) on pro-environmental behaviour.

Research on pro-environmental behaviour mainly focuses on agriculture, psychology, tourism, and business management, and there is a lack of research in the field of education, especially among young people. In terms of study design, current research focuses on the factors that influence pro-environmental behaviour, but rarely on those that predict it. In terms of assessment tools, research has focused on environmental attitude scales and observational methods, while research on the use of decision trees to predict pro-environmental behaviour has been rare. Decision-tree models, one of the data-mining algorithms in machine learning, have high predictive accuracy and the ability to decompose a complex decision process into a series of simpler decisions, thus providing a more easily interpretable solution [10,11,12]. Researchers have used decision-tree models to accurately predict the factors influencing the success and failure of innovation in the Korean manufacturing industry [13]. Other researchers have utilised decision-tree models in marketing and psychology to predict the response rate of consumer satisfaction, attitude, and loyalty surveys [14]. Therefore, the aim of this study was to explore the validity of applying decision-tree models to predict pro-environmental behaviour and to determine which variables could be used as predictors of pro-environmental behaviour.

## 2. Review

### 2.1. Pro-Environmental Behaviour

As early as 1990, Hines argued that pro-environmental behaviour is conscious and is guided by personal attitudes and personal responsibility [15]. Most scholars and researchers agree that this behaviour can be defined as people’s conscious efforts to reduce the negative impact of their personal behaviour on the natural environment [16,17,18]. However, other scholars regard pro-environmental behaviour as the result of a decision and preference [19]. Some researchers have distinguished pro-environmental behaviour into behaviour in the private sphere and behaviour in the public sphere. Furthermore, Steg et al. (2014) defined pro-environmental behaviour as a set of actions to improve the quality of the environment and promote sustainable development [20]. In this article, pro-environmental behaviour is defined as a conscious effort by college students to reduce the negative effects of their personal behaviour on the natural environment and the actions and lifestyles to promote environmental sustainability.

### 2.2. Measurement and Models of Pro-Environmental Behaviour

Currently, pro-environmental behaviour is usually measured by scales. Researchers often use or adapt previously developed scales as questionnaire instruments, for example, the former Environmental Attitudes Scale [21]. Blok et al. developed the Environmental Awareness Scale based on Steg (1999) [22].

Pro-environmental behaviour is also analysed in case studies. Researchers have also used observation and interviews to analyse pro-environmental behaviour in people’s everyday behaviour in specific settings. For example, researchers have used case studies to examine the process of pro-environmental behaviour change [23]. Examples of such case studies include studying energy-saving behaviour in two buildings to identify pro-environmental behaviour [24], measuring pro-environmental behaviour in terms of reducing household food waste [25], and observing behaviour that people use lifts and turn off lights to see if automated technology disrupts the development of pro-environmental behaviours [26].

Common models of pro-environmental behaviour are drawn from three main theoretical frameworks. The first is the theory of planned behaviour. This theory assumes that behaviour is determined by human motivation and that behavioural intentions are in turn the result of attitudes toward behaviour, subjective norms, and perceived behavioural control [27,28]. Thus, it can be seen that human behavioural intention is a key predictor of behaviour. Some researchers have constructed models based on this theory to investigate pro-environmental intentions and pro-environmental behaviour in tourism [19] and to predict pro-environmental behaviour in the workplace [22]. In addition, Schwartz proposed a norm-activation model of pro-environmental behaviour in 1977 and suggested that a sense of moral obligation or “personal norms” influences environmentally friendly intentions and behaviours [29]. Moreover, the values–beliefs–norms model was developed by Stern on the basis of value theory, normative activation theory and the new ecological theory (Unsworth, Dmitri, 1977) [30]. According to this model, human values are related to beliefs, and those beliefs shape individual behaviours through norms. This model reveals that individual values, beliefs and norms might influence pro-environmental behaviour [8]. According to this theory, pro-environmental behaviour is the result of pro-social norms, which are the result of certain beliefs (for example, an ecological worldview). Researchers have predicted the pro-environmental behaviour of tourists when visiting tourist attractions based on the value-belief-norm theory [8].

### 2.3. Methods for Predictive Analysis of Pro-Environmental Behaviour

The main predictive analysis methods for pro-environmental behaviour are regression analysis, validating factor analysis (CFA), meta-analysis, and principal component analysis. Applying CFA and structural equation modelling to the data, Trivedi et al. revealed that environmental control points and pro-environmental behaviour predicted consumer willingness to pay for green products [31]. Wesselink et al. used principal component analysis to suggest that leadership behaviour (as exemplary behaviour) and organisational support for the environment influenced employees’ pro-environmental intentions and behaviour [32]. Researchers have also applied multivariate logistic regression models to predict explanatory pro-environmental behaviours, proposing that specific preferences are important in predicting specific pro-environmental behaviours [19,33]. However, few researchers have used decision-tree models to predict pro-environmental behaviour [34].

### 2.4. Predictors of Pro-Environmental Behaviour

In the existing academic literatures, few studies focus on college students’ pro-environmental behaviour and very few studies use decision trees to predict pro-environmental behaviour. Therefore, the purpose of this study is to investigate: How can decision-tree models be used to predict college students’ pro-environmental behaviour? From the literature, we identified several predictors of pro-environmental behaviour (See Figure 1).

First of all, this study proposes perceived behavioural control and subjective norms as predictors for pro-environmental behaviour. Subjective norms are individuals’ perceptions of social pressure, as evidenced by their consideration of whether significant others will approve of their behaviour [35]. Some research findings suggest that subjective norms, perceived behavioural control, and place attachment have a positive effect on travellers’ attitudes toward environmentally responsible behaviour [36].

Furthermore, there is a linkage between pro-environmental social identity and pro-environmental behaviour. Social identity refers to an individual’s identification and evaluation based on their membership in social groups. One meta-analysis has demonstrated that pro-environmental social identity predicts collective pro-environmental action [37].

Moreover, this study suggests sense of place as a predictor for pro-environmental behaviour. Sense of place, the connection between individual feelings and place [38], has been divided into three dimensions: place identity, place dependence, and place attachment [39]. Daryanto and Song (2021)’s research reveals that individuals’ place attachment can lead to pro-environmental behaviour [34]. For example, individuals’ attachment to a sporting venue that they frequently visited can promote their pro-environmental behaviour at that venue.

Most importantly, the willingness to behave in an environmentally responsible manner, environmental activism, and innovative behaviour might be effective predictors for pro-environmental behaviour. A few studies reveal that the willingness to behave in an environmentally responsible manner [40], environmental activism [41], and innovative behaviour [42] can lead to pro-environmental behaviour. In addition, innovation is a driving force for the sustainable development of human society. Some studies have shown that technology is crucial for advancing sustainable development; examples include establishing normative learning channels, developing measures to address the interests of underserved populations, reforming systems and repositioning innovation systems [43]. It is important to explore the relationship between innovation capacity and pro-environmental behaviour.

Therefore, this study proposes several predictors for pro-environmental behaviour: subjective norms, perceived behavioural control, social identity, sense of place, the willingness to behave in an environmentally responsible manner, innovative behaviour, and environmentally responsible behaviour.

## 3. Materials and Methods

### 3.1. Participants

This study was conducted in Guangdong Province, China. Prior to finalising the research design, the researchers conducted exploratory focus interviews with five volunteer participants to identify possible predictive associations between pro-environmental behaviour and these proposed predictors including perceived behavioural control, social identity, innovative behaviour, sense of place, subjective norms, environmental activism, and the willingness to behave in an environmentally responsible manner.

The sampling process was divided into three steps. First, questionnaires were distributed to five university students from Guangdong Province. Second, after communicating with them individually, we selected other university students as the survey participants. All participants were informed of the purpose and content of the survey and agreed to participate in this survey. In the end, 336 students participated and completed the questionnaire. After data collection was completed, we determined that the actual number of valid questionnaires was 334. Of these, 68.56% of respondents were female and 31.44% were male.

### 3.2. Data Collection and Instruments

An online questionnaire was used as the data collection method. The questionnaire was completed from 28 September to 15 October 2021. Students scanned the quick-response (QR) code, accessed the fill-in screen for the questionnaire, answered the questions, and clicked on an icon to submit their response when they were finished (A QR code is a readable barcode that contains a lot of information. A device such as a mobile phone or tablet scans the QR code with a camera, recognises the binary data, and allows access to a specific link). Prior to scanning the code, respondents received details of the purpose of the study, and all completed the questionnaire on a voluntary basis.

The questionnaire consisted of demographic information and several scales. The demographic information included gender, age, school attended, discipline studied, and specialization The scales measured the following predictors of pro-environmental behaviour: perceived behavioural control, social identity, innovative behaviour, sense of place, subjective norms, environmental activism, and the willingness to behave in an environmentally responsible manner.

To improve the quality of the translation, the back translation method was used in this study, in which the first researcher translated the English into Chinese, then the second researcher back-translated the translated English into Chinese, and then the third researcher compared the original, translated and back-translated versions of the scale to finally assess the accuracy of the translation.

The research tools are shown in the following Table 1.

### 3.3. Reasons for Choosing a Decision-Tree Model

We chose a data-mining approach to process the data collected. Decision trees, machine learning algorithms, have proven to be suitable for predicting and revealing students’ academic achievement, psychological status, and other studies. We chose a decision-tree approach after considering the following aspects: (1) Decision-tree models have been widely used in studies on the prediction of students’ academic achievement and their psychological and behavioural perceptions [48,49,50,51,52,53]. (2) Decision-tree models are able to establish easily understandable classification rules with good interpretability. Such models output a top-down graph that expresses the rules. (3) Decision-tree models can handle complex relationships between predictor variables, demonstrated by its tolerance of multiple-covariance results. Therefore, we constructed a categorical decision-tree model to predict the level of pro-environmental behaviour of university students and to analyse the importance of each factor in predicting pro-environmental behaviour.

### 3.4. Construction of the Decision-Tree Model

The decision-tree model requires the classification of samples based on the information entropy of the input data set, which reflects the internal complexity of the sample. The greater the complexity and volatility within the sample, the greater the value for information entropy. Where *D* is the training data set with sample size *m* and is the probability of each class of samples [54].
(1)$$Entropy\left(D\right)=-\sum_{k=1}^{m}P_{k}{log}_{2}P_{k}$$

We chose the C5.0 algorithm, which uses information gain as a feature for judging the predictor variables. In this study, we divided the data collected sample into training samples and test samples. The information gain ratio is used to measure the difference in information entropy of the dataset under different classification methods. When the variable *C* is chosen to divide the dataset *D* into *n* subsets, the information gain ratio is defined as [55].
(2)$$Gain\;ratio\;(D,\;C)=\frac{Entropy\left(D\right)-Entropy(D|C)}{Entropy\left(C\right)}$$

#### 3.4.1. Pruning of the Decision-Tree Model

Based on the decision-tree model constructed from the training samples, the dataset was recursively started to each leaf node; that is, the leaf nodes were pruned layer by layer using the post-pruning method. The decision tree model constructed based on the training samples starts recursion of the dataset to each leaf node, which means that the leaf nodes are pruned layer by layer using a post-pruning method [56].

#### 3.4.2. Evaluation of the Decision-Tree Model

Of the sample data, 68% (*n* = 227) were selected as training data and 32% (*n* = 107) as test data. Accuracy, precision, and recall were the metrics we used to evaluate the quality of the model. Where accuracy is the proportion of correctly classified samples to all samples. We defined accuracy as the proportion of true positive samples to the proportion of samples with positive prediction results. We defined recall as the proportion of correctly predicted positive samples to realistic samples.

### 3.5. Data Analysis

We applied SPSS 23.0 to the data to obtain descriptive statistics and Modeler 18.0 to analyse the decision-tree model. First, descriptive statistical analysis was used to analyse frequency statistics, changes in correlation trends, which act as measures of high and low levels of pro-environmental behaviour. Second, we used the C50 algorithm to construct the decision-tree-analysis model and to determine which variables predicted the production of pro-environmental behaviour.

#### Coding of Key Variables

The sample was divided into two levels of pro-environmental behaviour, high and low. The questionnaire was scored on a 5-point Likert scale, and we selected 60% as the middle node for the study. The key variables for predicting pro-environmental behaviour were coded in this study based on the above principles (see Table 2).

## 4. Results

### 4.1. Descriptive and Correlation Analysis

Table 3 reports the mean and standard deviation of each predictor variable. The mean shows whether a predictor variable can be classified as high degree behaviour, or low degree behaviour, on average. The variance and standard deviation show the fluctuations in the scores for a predictor variable. Except for social identity, the mean for all predictor variables was greater than 3. Among the variances and standard deviations, the equal scores for social identity fluctuated less. This indicates that most participating university students are aware of knowledge related to the environment, but have some difficulty in relating it to their social life.

The correlations of the variables were assessed by Pearson’s product difference correlation coefficient and the results are presented in Table 4. Innovative behaviour, sense of place, subjective norms, environmental activism, willingness to behave in an environmentally responsible manner and perceived behavioural control were significantly and positively correlated with pro-environmental behaviour.

### 4.2. Predictive Analysis of Pro-Environmental Behaviour

According to Figure 2, the predictor variables for pro-environmental behaviour are willingness to behave in an environmentally responsible manner, innovative behaviour, and perceived behavioural control. The percentage of those with high pro-environmental behaviour was 63.88%. According to Figure 3, the importance of willingness to to behave in an environmentally responsible manner is 0.1562, innovative behaviour is 0.1404, and perceived behavioural control is 0.1322.

### 4.3. Evaluation of the Model

Table 5 and Table 6 show the confusion matrix and classification accuracy of the model, respectively. The precision of the model was 73.13% for the training sample and 69.16% for the test sample. According to Table 7, for the test sample, the model has a predictive accuracy of 72.89% and a recall of 90.87% for highly pro-environmental behaviour.

## 5. Discussion

This study uses a decision-tree model to predict pro-environmental behaviour, which is innovative in terms of choice of research design. It shows that the decision-tree model has some predictive validity. As previously stated, few studies have predicted pro-environmental behaviour. The results of this study showed that the decision-tree model was able to predict pro-environmental behaviour. In this study, using 60% as the threshold, the classification accuracy and recall of the model were greater than 60%, and the model showed a significant level of predictive accuracy of 72.89% for high pro-environmental behaviour, indicating that the predictive analysis in this study was effective.

Most importantly, a seven-factor model of pro-environmental behaviour is constructed by using the decision-tree. The predictor variables for pro-environmental behaviour are, in order of importance: the willingness to behave in an environmentally responsible manner, environmental activism, subjective norms, sense of place, innovative behaviour, social identity and perceived behavioural control. These findings are consistent with previous research.

First, willingness to behave in an environmentally responsible manner was the most important predictor variable, which is consistent with previous findings. Existing research has focused on the impact of corporate environmental responsibility and consumer environmental responsibility on pro-environmental behaviour [57]. In addition, one study has shown that employees show more pro-environmental behaviour at work when they recognise that their organisation is fulfilling or working toward corporate environmental responsibility [6]. Perceived corporate social responsibility can promote green consumption behaviour among customers. Customers’ emotions and their identification with the company are the mediating variables between CSR and green consumption behaviour. Some research findings suggest that environmental responsibility has a positive effect on people’s motivation to consume green products. Environmental awareness plays a partially mediating role between environmental responsibility and green consumption behaviour. Price sensitivity plays a negative moderating role in environmental responsibility, environmental awareness and green consumption behaviour [58].

Second, environmental activism was the second most important predictor variable. It manifests mainly in advocacy for research on behaviour concerning the environment, which is consistent with previous research. The environmental citizenship involved in pro-environmental behaviour can be linked to environmental activism through collective action, rather than through consumer behaviour and willingness to pay [41]. Automated technology may change people’s behaviour by reducing personal responsibility [26]. A study of two buildings in central London, in the United Kingdom, which included interviews with the head of energy efficiency, concluded that key actors had a positive effect and influence on promoting pro-environmental behaviour among employees [24].

Third, subjective norms, sense of place, and innovative behaviour were ranked as the t next most important predictor variables, which aligns with the results of previous research. In an investigation of the travel patterns of university students, findings suggested that effective regulations and positive social support, such as improving travel comfort, encourage people to choose more environmentally friendly travel patterns [27]. Correia and Sousa et al. studied the factors that influence pro-environmental behaviour of students in higher education. Their findings indicated that students’ subjective norms and perceived behavioural control had a positive impact on their pro-environmental intentions. Students’ perceived behavioural control and pro-environmental intentions had a strong and positive impact on their pro-environmental behaviour [59]. Local attachment has a positive impact on pro-environmental behaviour, and its impact is stronger in collectivist countries than in individualist countries [34]. Corporate social innovation, a novel strategic tool, combines innovative and environmental behaviour. One study found that training conducted by corporate universities significantly influenced employees’ innovative and environmental behaviours at work and in life, and also enhanced their normative commitment to the organisation [60].

Finally, based on the theory of planned behaviour, it is known that behaviour is determined by human motivation and that behavioural intentions are in turn the result of attitudes towards behaviour, subjective norms and perceived behavioural control. This theory can explain the predictive mechanisms of the elements’ pro-environmental behaviour in this study. The generation of decision tree nodes and the importance of elements can be demonstrated through previous research. The results of this study expand the perspective of predicting pro-environmental behaviour and validate previous research.

## 6. Significance

In a theoretical sense, this study uses machine learning to predict variables regarding pro-environmental behaviour, which deepens the understanding of what pro-environmental behaviour entails. In addition, we found that willingness to act responsibly, innovative behaviour, and perceived behavioural control have a role in influencing pro-environmental behaviour.

In a practical sense, the seven possible predictor variables proposed we proposed and the three predictor variables that emerged from the model can help researchers gain a deeper understanding of the mechanism in pro-environmental behaviour.

## 7. Limitations and Future Directions

There are some limitations to this study. First, this study used a cross-sectional design. Second, study participants were all from one university in Guangdong Province, so the study sample limits the generalisability of the findings. Future participants could adopt a longitudinal research design and recruit participants from different regions and institutions. In addition, future researchers could explore other variables that predict the pro-environmental behaviour of university students. Demographic factors can also be added to the decision tree model in the future research. For example, the gender of the participants, their disciplinary orientation and their propensity to be associated with secondary school arts and sciences, etc. Future research is also needed on how to enhance pro-environmental behaviour among university students from the perspective of the identified predictor variables, thus leading to increased sustainability of their quality of life.

## 8. Conclusions

Firstly, based on the results of the decision-tree model, it can be seen that the predictors of university students’ pro-environmental behaviour are in order of importance, willingness to behave in an environmentally responsible manner, environmental activism, subjective norms, sense of place, innovative behaviour, social identity, and perceived behavioural control.

Finally, the results of this modelling provides a certain level of support for using decision trees to predict university students pro-environmental behaviour.

## Figures and Tables

**Figure 1 ijerph-19-09407-f001:**
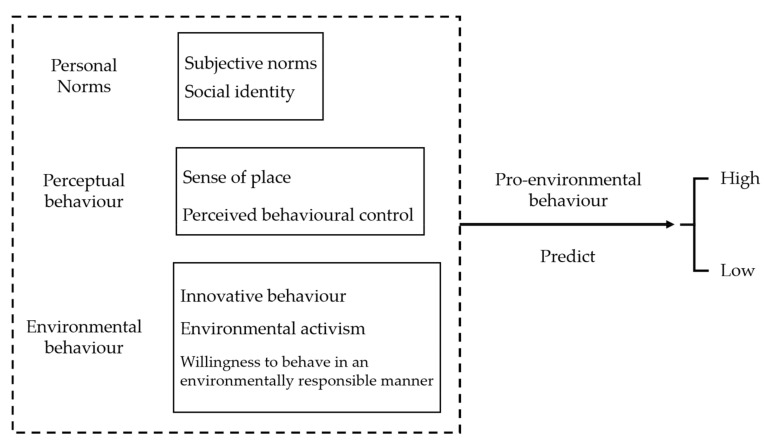
Hypothetical predictions of pro-environmental behaviour.

**Figure 2 ijerph-19-09407-f002:**
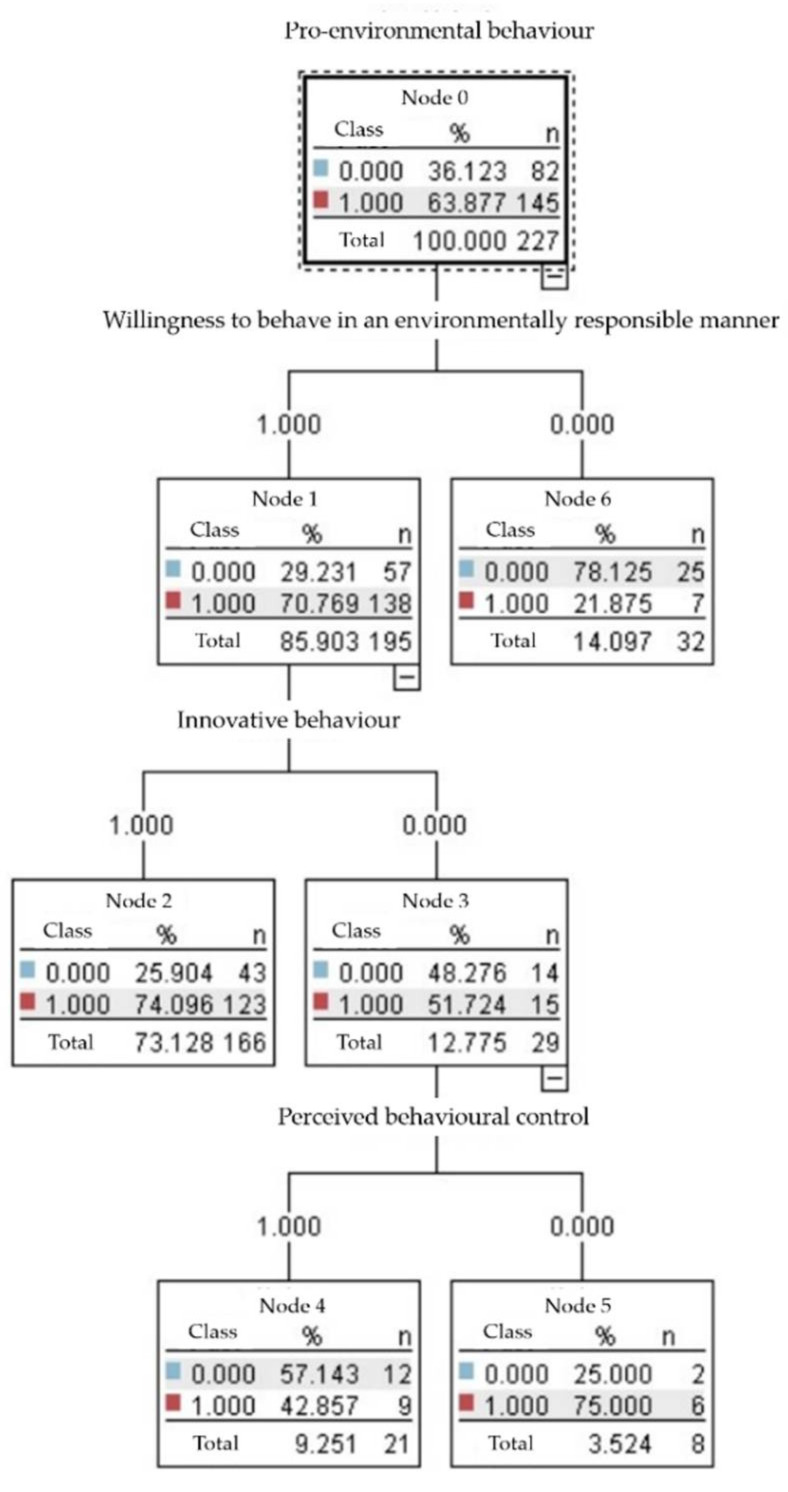
Predictive models for pro-environmental models.

**Figure 3 ijerph-19-09407-f003:**
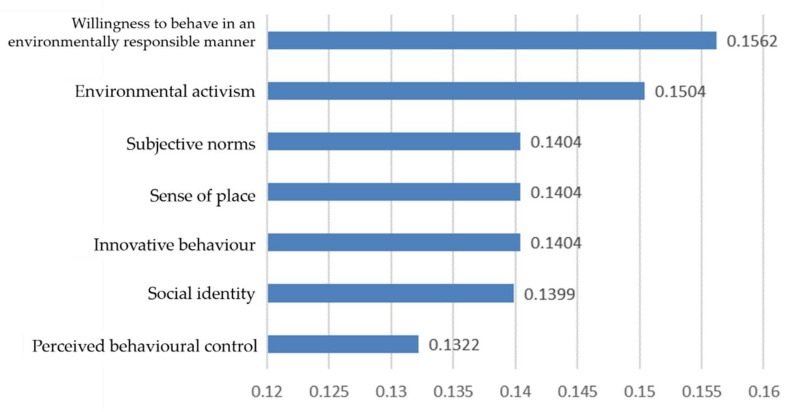
Predictor variables of pro-environmental behaviour.

**Table 1 ijerph-19-09407-t001:** Research instruments.

Serial Number	Tool Name	Provenance	Number of Items	Scoring Method	Sample	Reliability
1	Perceived Behavioural Control Test	[36]	3	Each item was rated on a five-point Likert scale: 1 (“completely disagree”), 2 (“some-what disagree”), 3 (“neutral”), 4 (“somewhat agree”), and 5 for (“completely agree”).	“My action is important for reducing environmental white pollution on campus” “My actions make a substantial contribution to protecting campus environment” “My action to protect the campus environment is not a waste of time or money”.	0.844
2	Social Identity Test	[44,45]	10		“I am a person who thinks the group is important” “I am a person who identifies with the group” “I am a person who feels a strong connection to the group”	0.654
3	Innovative Behaviour Test	[46]	9		“I often generate new ideas when I encounter difficulties”, “I seek out new ideas, techniques, or tools,” “I come up with original solutions to problems”	0.938
4	Sense of Place Test	[47]	12		“This place is closely associated but does not really define who I am”, “This place is so closely associated to me that I can be my true self”, “This place explains everything about who I am as a person”,	0.924
5	Subjective Norms Test	[36]	3		“Those who are important to me think I should take action to protect the campus envi-ronment”, “Those who are important to me would want me to take action to protect the campus environment”, and “Those who are important to me would be happy if I took action to protect the campus environment”.	0.926
6	Environmental Activism Test	[41]	6		“I participate in activities organised by environmental groups” “I give financial sup-port to an environmental group” “I circulate petitions asking for improvements in government environmental policies”	0.812
7	Test of willingness to behave in an environmentally responsible manner behaviour	[40]	6		“I will go to learn about environmental protection”, “I will remind my friends not to litter on campus”, “I will make a donation to support environmental protection on campus”,	0.897
8	Test of Pro-environmental Behaviour	[40]	14		“I will make a special effort to buy pesticide-free fruits and vegetables once”, “I would consider myself a member of any group whose main goal is to protect the environment “, “In the last 12 months I have read newsletters, magazines or other publications written by environmental organizations”.	0.902

**Table 2 ijerph-19-09407-t002:** Coding of variables.

Variables	Code	Number	Percentage
Pro-environmental behaviour	0 = Low	115	34.43%
	1 = High	219	65.57%
Gender	0 = Low	229	68.56%
	1 = High	105	31.44%
Perceived behavioural control	0 = Low	77	23.05%
	1 = High	257	76.95%
Social identity	0 = Low	243	72.75%
	1 = High	91	27.26%
Innovative behaviour	0 = Low	77	23.05%
	1 = High	257	76.95%
Sense of place	0 = Low	75	22.46%
	1 = High	259	77.54%
Subjective norms	0 = Low	85	25.45%
	1 = High	249	74.55%
Environmental activism	0 = Low	44	13.17%
	1 = High	290	86.82%
Willingness to behave in an environmentally responsible manner	0 = Low	47	14.07%
	1 = High	287	85.93%

**Table 3 ijerph-19-09407-t003:** Descriptive statistics.

Variables	Maximum	M	Variance	SD	60% of Maximum
Perceived behavioural control	5	3.83	0.5	0.71	3
Social identity	5	2.87	0.15	0.39	3
Innovative behaviour	5	3.71	0.38	0.62	3
Sense of place	5	3.59	0.38	0.62	3
Subjective norms	5	3.84	0.53	0.73	3
Environmental activism	5	3.73	0.32	0.57	3
Willingness to behave in an environmentally responsible manner	5	3.91	0.42	0.65	3
Pro-environmental behaviour	5	3.34	0.39	0.63	3

**Table 4 ijerph-19-09407-t004:** Pearson’s r of the variables.

Variables	SI	IB	SP	SN	EA	WBERM	PBC	PEB
SI	1							
IB	0.111 *	1						
SP	0.118 *	0.677 **	1					
SN	0.137 *	0.614 **	0.685 **	1				
EA	0.054	0.463 **	0.521 **	0.558 **	1			
WBERM	0.110 *	0.591 **	0.624 **	0.729 **	0.570 **	1		
PBC	0.159 **	0.584 **	0.687 **	0.728 **	0.513 **	0.729 **	1	
PEB	0.032	0.478 **	0.535 **	0.500 **	0.640 **	0.515 **	0.465 **	1

Note: SI = Social identity, IB = Innovative behaviour, SP = Sense of place, SN = Subjective norms, EA = Environmental activism, WBERM = Willingness to behave in an environmentally responsible manner, PBC = Perceived behavioural control, PEB = Pro-environmental behaviour * *p* < 0.05, ** *p* < 0.01.

**Table 5 ijerph-19-09407-t005:** Confusion matrix.

		Predicted Class	
		Class = Low	Class = High
Actual class of training data	Class = Low	31	51
	Class = High	10	135
Actual class of testing data	Class = Low	10	23
	Class = High	10	64

**Table 6 ijerph-19-09407-t006:** Classification accuracy.

Title 1	Title 2	Number	Proportion
Training data	Correct	166	73.13%
	Wrong	61	26.87%
	Total	227	
Testing data	Correct	74	69.16%
	Wrong	33	30.84%
	Total	107	

**Table 7 ijerph-19-09407-t007:** Recall and precision of the prediction model.

	Recall Rate ^1^	Precision Rate ^2^
Low pro-environmental behaviour	35.65%	67.21%
High pro-social behaviour	90.87%	72.89%

^1^ Recall = TP (true positive) divided by TP (true positive) plus FN (false negative). ^2^ Accuracy = TP (true positive) divided by TP (true positive) plus FP (false positive).

## Data Availability

Not applicable.

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
