# Peer review of "Predictive Analysis of the Pro-Environmental Behaviour of College Students Using a Decision-Tree Model"

_ijerph, 2022, doi:10.3390/ijerph19159407_

Round 1
Reviewer 1 Report
This paper uses a decision-tree model to explore potential predictors of pro-environmental behavior among university students in China. The authors find that willingness to behave in an environmental responsible manner was most closely related to pro-environmental behavior, and a number of other variables were also related to behavior including environmental behaviorism, creative behavior, and perceived behavioral efficacy. The overall approach to the paper may provide some value to the literature given the focus on decision-tree models, but the paper is unclear in a number of places that limits what can really be learned from the present study.
Strengths
1. Use of decision-tree models in the pro-environmental behavior context is fairly limited, so this paper could help demonstrate the value in this type of modeling for understanding pro-environmental behavior.
2. The authors have used an interesting selection of potential predictors of pro-environmental behavior that is not always explored in the same study (e.g., willingness, creativity, social identity, sense of place).
Limitations
1. The introduction feels like it is trying to cover a very wide range of studies related to pro-environmental behavior, and thus feels a little disorganized and perhaps more inclusive than it needs to be. I think focusing a bit more on meta-analytic findings in the area could help organize the introduction by showing what previous systematic reviews have found to be linked to pro-environmental behaviors. Alternatively, the authors may want to hone in on the variables they chose to explore in their study, and not spend much time summarizing research on other potential predictors, which would help hone in on the literature most relevant to their proposed research.
For example, the authors discuss the relevance of attitudes for predicting pro-environmental behavior (p. 2), but attitudes are not strictly speaking a measure used in this study. I personally would not consider measures of norms, social identity, sense of place, or efficacy to be measures of attitudes, and thus would not spend a lot of time talking about attitudes specifically in the introduction.
Another example is discussing intrinsic motivation and moral obligations in the discussion on p. 11 in the paragraph focusing on willingness to behavior in an environmentally responsible manner. Are the authors saying that willingness is a form of intrinsic motivation? Or a moral motivation? Those seem like a stretch to me, and thus sort of combining all of those constructs when discussing the willingness results could just confuse the reader regarding the findings and how broadly they can be applied to other constructs. The same goes for discussion of self-affirmation practices in the next paragraph on the environmental behaviorism findings. I just struggle to see how those things are connected in the context of this study.
2. Relatedly, the authors are sometimes a little unclear in the language they use to describe measures. For example, creative behavior is sometimes called innovative behavior, and those two ideas may be closely related, but it would be helpful to stick with the most accurate term throughout the paper. Also, typically perceived behavioral control is the name of the construct used in studies of pro-environmental behavior, or some form of efficacy (i.e., self-efficacy and/or response efficacy). I unsuccessfully tried to track down the #38 citation by Zhou et al. (2014), but I am not familiar with perceived behavioral efficacy. Is that indeed the name of the construct used in the Zhou et al. (2014) paper?
3. It is a minor point, but I am not used to describing perceived behavioral control/efficacy as necessarily having an altruistic component (p. 4).
4. Perhaps the most unexpected variable included in the study is creative behavior. The authors cite one study related to business innovations and sustainability as background for why creative behavior was included in the study, but a little more discussion of why this variable was included/why we might expect creative behavior to be linked to pro-environmental behavior would be helpful.
5. The authors refer to their sample as adolescents, but it appears the sample is university students, which would be a very common type of sample found in previous literature. Is there something more unique/are the study participants younger than typical college samples? If they are on average 18+ years of age, I am not sure calling them adolescents is accurate, unless the authors have additional justification for that sample description.
6. Was there a need to translate any measures? If so, what was that process?
7. I apologize if I am just missing it, but I do not see any description of how the outcome variable of pro-environmental behavior was measured. This is particularly important given a couple of the predictors may be quite closely related to/similarly measured as the outcome, including willingness to behave in an environmentally responsible manner and environmental behaviorism. It might also be nice to include a correlation matrix to just demonstrate these three measures are not too highly correlated.
8. Did the prompt for the social identity measure define “the group” for participants? Were participants supposed to be thinking about other university students, or some other group? Or just “the group” however the participant chose to define it?
9. Is environmental behaviorism the best name for that scale? It seems to mostly focus on environmental advocacy/activism. The name environmental behaviorism suggests to me a broader set of environmental actions (e.g., saving energy, taking public transportation, recycling, eating less meat). Likewise, the willingness measure seems to mostly focus on advocacy as well.
10. Did the authors consider including demographic measures in the decision-tree model? It may be worth including them in a version of the model.
11. It would be useful if the authors could provide more explanation for why the results in Figure 2 and Figure 3 vary, as the authors seem to sometimes describe the most important/strongest predictors of pro-environmental behavior as deriving from the analyses related to Figure 2, and elsewhere in the paper as deriving from the analyses related to Figure 3. However, these two sets of results do differ a bit, so more explanation for how to think about these two sets of results would be helpful.
The lack of clarity regarding these two sets of analyses is most apparent in the first two paragraphs of the conclusion. I cannot really tell the difference in what these two paragraphs are trying to communicate, even though the order of the importance of the predictors is obviously different.
Author Response
We would like to thank you for your helpful comments and constructive suggestions, which has significantly improved the presentation of our manuscript. We have carefully considered all comments from the reviewers and revised our manuscript accordingly. In the following section, we response to each comment from the reviewers. We believe that our responses have well addressed all concerns from the reviewers.
1.Thank you for your suggestions. In our revisions, we have added several literatures on meta-analytic findings in the area of pro-environmental behavior and focusing.
See lines 50-61.
2.Thank you for your valuable suggestion. We fully agree that norms, social identity, sense of place, or efficacy are not related to measures of attitudes. The discussion will focus on the relationship between social identity, norms and sense of place with pro-environmental behaviors.
- I could not agree with you more. We fully agree that the willingness to behave in an environmentally responsible way is neither intrinsic motivation nor moral motivation. We focus on the impact of the willingness to behave in an environmentally responsible way on pro-environmental behaviour. See line 419-432
- Thank you so much for your careful check. We stick with the most accurate term of innovative behavior and perceived behavioral control throughout the paper.
- Thank you for your suggestion. We can’t agree with you more.
- Thanks for your suggestions. We have added a few examples from researchers, academics and UNESCO to highlight why individuals' transformative and creative actions are crucial for sustainability.
See line 207-219.
- Thank you very much for your suggestion. As the sample is the first-year university student instead of adolescents (age 10-19), we use youth (age 15-24) here.
- Thanks for your suggestions. We have added a back translation method to finally ensure the accuracy of the translation.
- Thank you for your suggestion. We have added the description of the measurement of outcome variable of pro-environmental behavior.
- Thanks for your suggestions. The participants are first-year university students.
- Thank for your suggestion. we use Environmental Activism Scale developed by Dono et al in this study.
- Thanks for your suggestions. We totally understand the reviewer's concern. This is the future direction of research. In our follow-up study, we will try to add demographic factors to the decision tree model.
- Thank you for your suggestion. We had added more explanation in the discussion section.
- Thanks for your suggestions. We have re-written this part according to the suggestion.
Reviewer 2 Report
The paper covers interesting research on predictive analysis of the pro-environmental behaviour of adolescents using a decision-tree model. Three hundred thirty-four university students in Guangdong Province, China, completed a questionnaire that consisted of seven parts: the Perceived Behavioural Efficacy Scale, the Social Identity Scale, the Creative Behaviour Scale, the Sense of Place Scale, the Subjective Norms Scale, the Environmental Behaviourism Scale, and the willingness to behave in an environmentally responsible manner Scale. This study establishes a link between machine learning and pro-environmental behaviour and broadens our understanding of pro-environmental behaviour. It provides research support for improving people’s sustainable development philosophy and behaviour. However, the manuscript is written in style more like a report rather than a research article. The global innovativeness in research development hasn't been presented. Some figures and tables which involve world-wide novel research should be described and discussed with more details to emphasize the state-of-the-art-review all over the world novelty. Please use this the newest (2018-2022) Web of Science journal papers.
Author Response
Thank you for your suggestion. I have added the literature reviews on the newest (2018-2022) Web of Science journal papers, which might increase their academic significance. We have also changed the descriptive language style and made it more analytical and critical. See lines 174-183.
Reviewer 3 Report
I consider the research paper very interesting, nevertheless there are some issues that need to be improved, namely:
- In line 17, there is a mistake in the written English : where it is: “and. UNESCO “, should be: “and UNESCO”
- In the abstract , where is: “: the Perceived Behavioural Efficacy Scale, the Social Identity Scale, the Creative Behaviour Scale, the Sense of Place Scale, the Subjective Norms Scale, the Environmental Behaviourism Scale, and the willingness to behave in an environmentally responsible manner Scale”, perhaps the last sentence should also be written I capital letters : “the Perceived Behavioural Efficacy Scale, the Social Identity Scale, the Creative Behaviour Scale, the Sense of Place Scale, the Subjective Norms Scale, the Environmental Behaviourism Scale, and the Willingness to Behave in an Environmentally Responsible Manner Scale”. Another hypotheses is not writing all the sentences in capital letters.
- In the section 3.2, I think the authors could summarize all the information- from 3.2.1. to 3.2.7 in an interesting table with all the scales and respective items and sources for each item.
- The research paper should use more references, particularly in the introduction and discussion sections. Some key references in the area, namely: correia et al., sousa et al., among others should be referred in this paper
- The authors should also improve the written English.
Author Response
- Thank you so much for your careful check. We have corrected the mistake.
-
Thanks for your suggestions. We are very sorry for our negligence in spelling. we have corrected the spelling and polished the language.
- Thanks for your suggestions. We have converted the contents of subsection 3.2 into a table. Please see table1.
- Thanks for your suggestions. We have added the literature reviews from these key academics, including Correia and Sousa.
- Thanks for your suggestions. We are making great endeavors to improve written English. We also seek suggestions and learn from professionals in the field of education and psychology.
Round 2
Reviewer 1 Report
This paper uses a decision-tree model to explore potential predictors of pro-environmental behavior among university students in China. The authors have updated the manuscript with suggestions from the reviewers, and I feel the paper is stronger because of these edits. However, I still have a couple of questions about the contents.
1. The authors have changed how they refer to the sample, formerly referring to adolescents and now referring to youth. The authors state in the paper that there is a lack of research on youth pro-environmental behavior. Even if the language has changed a little, the underlying issue may still remain, which is the fact that studies of pro-environmental behavior with college student samples are quite common. Unless there is something still unique about this specific sample, then I would not classify a college student sample as a particular strength or unique feature of the study. For example, what is the mean age of the sample? Is it under 18 years of age?
Additionally, if the authors are going to stick with the adolescents/youth description of the sample, they should provide citations to justify why this sample, with its mean age, should be considered adolescents/youth. I fear that people reading the paper will be surprised to learn these are college students given the paper title and abstract.
2. I appreciate that the authors have now included information about the outcome variable of pro-environmental behavior. However, it is not included in the new Tables 2 and 3. Also, even though 4.1 refers to correlation analysis, there still does not appear to be a correlation table in the paper. Given how similar some of the constructs are, it would be helpful to have a correlation table with all of the variables in it (including the outcome variable).
Additionally, does the outcome measure only focus on purchasing behaviors? The three examples in Table 2 are all purchasing behaviors (or things people are choosing to not purchase), so if all of the items for that measure focus on purchasing behaviors then it might make sense to relabel that variable as “pro-environmental purchasing” or something similar.
Author Response
Thank you for your helpful comments and constructive suggestions. We have carefully considered all comments from the reviewers and revised our manuscript accordingly. In the following section, we summarize our responses to each comment from the reviewers.
- Thank you very much for your suggestion. As the sample is college students. We used college students instead. We have made updates to the manuscript.
-
Thank you very much for your suggestion. We have updated the pro-environmental behavior in Tables 2 and 3. Thank you for your reminder. We have added a correlation analysis table to the manuscript. Please see Table 4.
- Thank you very much for your suggestion. We are very sorry for our negligence. All of the items for the measure don’t focus on purchasing behaviors. Other items for example, "I would consider myself a member of any group whose main goal is to protect the environment ", "In the last 12 months I have read newsletters, magazines or other publications written by environmental organizations", "In the last 12 months I have signed petitions in support of protecting the environment".
Reviewer 3 Report
Thank you for all the improovements.
Author Response
Thank you for your appreciation. We will use the editing services at MDPI.